# Fountain Pen-Inspired 3D Colloidal Assembly, Consisting of Metallic Nanoparticles on a Femtoliter Scale

**DOI:** 10.3390/nano13172403

**Published:** 2023-08-24

**Authors:** Sung-Jo Kim, Il-Hyun Lee, Won-Geun Kim, Yoon-Hwae Hwang, Jin-Woo Oh

**Affiliations:** 1BIT Fusion Technology Center, Pusan National University, Busan 46241, Republic of Korea; sungjokim84@pusan.ac.kr (S.-J.K.); kim1guen@postech.ac.kr (W.-G.K.); 2Department of Nano Fusion Technology, Pusan National University, Busan 46241, Republic of Korea; 2023710262@skku.edu; 3Department of Nano Engineering, SKKU Advanced Institute of Nanotechnology (SAINT), Sungkyunkwan University (SKKU), Suwon 16419, Republic of Korea; 4Department of Nano Science and Technology, SKKU Advanced Institute of Nanotechnology (SAINT), Sungkyunkwan University (SKKU), Suwon 16419, Republic of Korea; 5Department of Mechanical Engineering, Pohang University of Science and Technology (POSTECH), Pohang 37673, Republic of Korea; 6Department of Nano Energy Engineering, Pusan National University, Busan 46241, Republic of Korea

**Keywords:** colloidal assembly, nanoparticles, 3D nano-cluster, localized surface plasmon resonance, small structure

## Abstract

The 3D colloidal assemblies composed of nanoparticles (NPs) are closely associated with optical properties such as photonic crystals, localized surface plasmon resonance, and surface-enhanced Raman scattering. However, research on their fabrication remains insufficient. Here, the femtoliter volume of a 3D colloidal assembly is shown, using the evaporation of a fine fountain pen. A nano-fountain pen (NPF) with a micrometer-level tip inner diameter was adopted for the fine evaporation control of the ink solvent. The picoliters of the evaporation occurring at the NFP tip and femtoliter volume of the 3D colloidal assembly were analyzed using a diffusion equation. The shape of the 3D colloidal assembly was dependent on the evaporation regarding the accumulation time and tip size, and they exhibited random close packing. Using gold-, silver-, and platinum-NPs and mixing ratios of them, diverse 3D colloidal assemblies were formed. The spectra regarding a localized surface plasmon resonance of them were changed according to composition and mixing ratio. We expect that this could be widely applied as a simple fabrication tool in order to explore complex metamaterials constructed of nanoparticles, as this method is highly flexible in varying the shape as well as composition ratio of self-assembled structures.

## 1. Introduction

Small objects such as nanoparticles (NPs) have gained interest due to their utility in localized surface plasmon resonance (LSPR) [1,2,3,4,5]. The incident light to subwavelength scales of metallic NPs exhibits LSPR which is a useful optical phenomenon in diverse practical applications such as biosensors [4,5,6], energy devices [7,8,9], chemical synthesis [10,11,12] and surface-enhanced Raman spectroscopy [13,14,15]. To achieve the fabrication of high-performance functional optical devices, it is necessary to be able to design and manipulate LSPRs. Through the assembly of metallic NPs, plasmonic modes were able to be induced in individual NPs [16,17]. In the superradiant mode in which these dipole plasmons vibrate in phase, the radiation damping increases and the spectrum broadens, and when they vibrate out-of-phase, the radiation damping decreases, causing a spectral dip. These ensembles of plasmon modes depend on the NPs cluster and determine the LSPR characteristics. The plasmon modes related to NPs cluster can be tuned by the shape, size, dielectric surroundings, and interparticle distances of NPs [16,17], and this tunability opens the possibility for sensor applications. Recently, it has also been demonstrated that LSPR can be enhanced or manipulated by changing the geometrical packing structure or composition of NPs clusters [18,19,20,21].

The lithography with top-down fabrication has long been adopted to develop plasmon structures [16,17,22,23]. Electron- or ion-beam lithography is capable of fabricating nanometer-sized metallic objects by adjusting the shapes and gap distances; however, it suffers from cost inefficiency and low throughput. In addition, the fabricated structures are restricted to two-dimensions, and expanding to three-dimensions through vertical stacking is challenging. Using colloidal assembly has emerged as an alternative method to make 3D plasmon clusters instead of top-down fabrication [24,25,26]. This method utilizing solution-processed self-assembly provides advantages in terms of simplicity, scalable, and high-throughput route for fabricating 2D as well as 3D plasmonic clusters. Marangoni flow caused by nonuniform evaporation-induced surface tension gradient contributes to the assembly of NPs resulting in densely packed clusters at a two- or three-phase interface, and this solution-mediated mass transfer process such as dip coating [24], the Langmuir–Blodgett process [25], and template-assisted self-assembly [26] have been devised for fabricating NP clusters. Although many colloidal assembly techniques have been developed, constraints on the free 3D structure creation and the change in component composition ratio of plasmon clusters consisting of NPs remain challenging issues.

In this work, we propose an evaporation-induced fabrication and working mechanism for the 3D colloidal assembly consisting of metallic NPs. In order to precisely control the evaporation of the ink solvent, a micrometer-scale fountain pen tip is introduced into the 3D colloidal assembly process. Nanoparticles dispersed in ink using picoliter volume level evaporation are used to fabricate a 3D NP cluster with a femtoliter volume at the micrometer-scale level, and the packing structure is analyzed. Finally, we investigate the LSPR properties of a variety of NP clusters created using NPs composed of gold, silver, and platinum.

## 2. Materials and Methods

### 2.1. Fabrication of Nano-Fountain Pen

To make a nano-fountain pen (NFP), which has an inner tip diameter of several micrometer levels, we used a micropipette puller (SU-P97 FLAMING/BROWN PIPETTE PULLER, Sarasota, FL, USA), where a standard glass capillary tube consisting of borosilicate glass (ITEM No. 1B100F-6, outer diameter = 1.0 mm) was used to fabricate NFP. These were purchased from World Precision Instruments. We adjusted the pulling conditions of the puller to 520–530${\;}^{\circ }C$ and 10–11 levels of pulling velocity, where the pulling velocity is a set unit used in the pulling machine representing the rate of separation of the puller bars when the glass capillary tube starts to melt. In this condition, the part of the melting capillary tube was pulled and changed to the hourglass neck. In the end, the neck is broken to create two NFPs with a narrow inner diameter tip ranging between 3–5μm.

### 2.2. Preparation of Nanoparticle-Dispersed Ink

The ink inserted into the NFP used for 3D colloidal assembly was a colloid consisting of liquid water (dispersion medium) and metallic NPs (dispersed phase). Metallic nanoparticles composed of gold (Au), silver (Ag), and platinum (Pt), respectively, were used as building blocks for the 3D colloidal assembly, and these nanoparticles were purchased from Nanocomposix (San Diego, CA, USA) and coated with polyvinylpyrrolidone (PVP) for dispersion. The diameters (*D*) of metallic NPs used are as follows; AuNPs (77 ± 9 nm and 20 ± 2 nm), AgNPs (71 ± 7 nm and 26 ± 4 nm), and PtNP (72 ± 4 nm). Using the mean diameters of NPs, we assigned the symbols to $G\equiv {\mathrm{AuNP}}_{77}$, $g\equiv {\mathrm{AuNP}}_{20}$, $S\equiv {\mathrm{AgNP}}_{71}$, $s\equiv {\mathrm{AgNP}}_{26}$, and $P\equiv {\mathrm{PtNP}}_{72}$ where uppercase and lowercase symbols represent relatively large and small nanoparticles. The concentrations (*c* in the unit of mg/mL) of the nanoparticles dispersed in the ink were defined as $c_{G}$, $c_{g}$, $c_{S}$, $c_{s}$, and $c_{P}$ according to the sizes and compositions of nanoparticles. In the case of the number density ($n=\frac{6c}{\pi D^{3}\tilde{\rho }}$, $\tilde{\rho }$ is a mass density of nanoparticle depending on the composition), we defined to $n_{G}$, $n_{g}$, $n_{S}$, $n_{s}$, and $n_{P}$.

### 2.3. Motorized Printing Process

Adjusting the 3D colloidal assembly was controlled by distance changes between the tip and the substrate using a 3-axis motorized stage operated by a controller (XPS-D4, Newport, Irvine, CA, USA), and the position of the initial NFP is adjusted using a 3-axis manual stage (PT3A/M, Thorlabs, Newton, NJ, USA). The height change on the *z*-axis of the NFP and substrate (Si wafer mounted on the 3-axis motorized stage) was measured using a side-view optical microscopy system consisting of a halogen fiber illuminator (OSL2, Thorlabs). Three steps were set for the stamping process, Accumulation–Contact–Detachment, while observing using an objective lens (MY20X-804, Mitutoyo, Kawasaki, Japan), and a charge-coupled device (CCD) camera. After a stamping process, moving the substrate along the *x*- or *y*-axis enables the formation of a regular pattern of 3D colloidal assembly structures on the substrate.

### 2.4. Structure and Optical Spectra of 3D Colloidal Assembly

The structure of the 3D colloidal assembly was measured using a scanning electron microscope (JSM-7900F, JEOL Ltd., Tokyo, Japan). We manually marked the center of mass of the NPs composing the structure using ImageJ software and found their center coordinates using the function of Analyze Particles. To investigate the optical properties of 3D colloidal assemblies, we used an optical microscope (BX53M, Olympus, Tokyo, Japan) in dark-field mode using an objective lens (LMPlanFL N, 100×, NA = 0.8, Olympus). The spectral characteristics of the scattered light collected by the microscope were measured with a spectrometer mounted on the microscope (US/USB4000, Ocean Optics, Duiven, The Netherlands), and the color from the measured spectra was evaluated using CIE 1931 color space plugged in Origin 2021.

## 3. Results and Discussion

### 3.1. 3D Colloidal Assembly Using a Nano-Fountain Pen

Water evaporation is ubiquitous, but occurs too slowly for us to perceive it visually. In particular, controlling the evaporation of liquid droplets in the aerosol state is a field of great interest as it occupies a very important part of engine technology, where the time required for complete evaporation is proportional to the radius (*R*) of a spherical droplet: this is known as the Radius-Square-Law [27]. At this time, the evaporating mass of liquid per unit of time from the surface of the droplet is proportional to *R*. When the liquid is continuously supplied to the droplet and *R* does not change, the mass loss is proportional to *R* and evaporation time ($t_{e}$), and a similar situation occurs at the tip of a fountain pen. In the ink inside the fountain pen, nanometer-level dye particles are dispersed in a solvent to express the color of the writing, and the nanoparticles gradually accumulate at the tip due to evaporation of the solvent occurring at the tip. If the tip with accumulated nanoparticles is stamped on a hard substrate such as glass where solvent absorption does not occur, the accumulated nanoparticles might move to the substrate leaving traces of a 3D colloidal assembly with the base area and height. The volume of the 3D colloidal assembly created through this stamping process is proportional to the inner diameter of the tip and the evaporation time. Since the scale of the 3D colloidal assembly’s base area is proportional to $R^{2}$, its height is proportional to $R^{-1}$ and $t_{e}$. Through this dimensional analysis, we reached the conclusion that it can be used to create the 3D colloidal assembly through the stamping process by reducing the size of the fountain pen tip or increasing the accumulation time. In particular, in order to prepare for the fast manufacturing speed for large-area processes in the future, it is more reasonable to reduce the inner diameter of the tip of the fountain pen than to increase the accumulation time. Therefore, we prepared a nano-fountain pen (NFP) with micrometer-sized levels of inner tip diameter for the 3D colloidal assembly (refer to Materials and Methods).

In order to precisely control the 3D colloidal assembly using the nano-fountain pen (NFP), it was important to control the evaporation of water in the ink that occurs at the NFP tip. The Si wafer was used as a substrate mounting on the 3-axis motorized stage (A of Figure 1a), and the motorized stage was utilized to adjust the stamping process of the 3D colloidal assembly. The 3-axis manual stage (B of Figure 1a) was used to set the initial position of the NFP hanging on the manual stage (C of Figure 1a). The motorized stage setting and assembly process was observed by the side-view microscope system that consists of the light source, objective lens, convex lens, and CCD camera (D–G of Figure 1a). We note that the stamping process setup was isolated from the external environment with an acrylic box to control humidity and temperature. In order to achieve 3D colloidal assembly using the NFP, the cartridge was filled with ink above a critical height (Figure 1b). The inner diameter of the NFP tip was $2R_{1}$, and the thick inner diameter of the ink cartridge of the NFP was $2R_{2}$, and $R_{2}$ was about 100 times larger than $R_{1}$. Following this, the ink volume filled in the NFP would be approximate to $V_{\mathrm{ink}}\approx \pi R_{2}^{2}H$ where *H* is the filling height of the ink in the cartridge. The ink filled in the NFP experiences three forces; Two capillary forces known as Young-Laplace Equation ($F_{1}=\pi R_{1}^{2}\times \frac{2\gamma \cos\theta }{R_{1}}$ and $F_{2}=\pi R_{2}^{2}\times \frac{2\gamma \cos\theta }{R_{2}}$) regarding the two water-air interfaces, and gravity $F_{3}=\rho gV_{\mathrm{ink}}$. Here, γ=72mN/m is the surface tension for the air-water interface [28], θ is the contact angle for the air-water-glass surface [29,30], ρ is the mass density of water, and *g* is the gravitational acceleration. So, the net force parallel to the *z*-axis is expressed as $F=-F_{1}+F_{2}-F_{3}$ and is approximate to $F\approx F_{2}-F_{3}$ due to $R_{2}\gg R_{1}$. In order for the protruding hemispherical ink droplets to form on the NFP tip, $F_{3}$ must be greater than $F_{2}$, and the height of ink that must be filled in the cartridge to meet the condition expressed as $H>H_{C}$, where a critical height, $H_{C}=\frac{2\gamma \cos\theta }{\rho gR_{2}}$ coincides with a well-known capillary rise of liquid in a capillary [28]. Therefore, the ink filled the cartridge over 17 mm for easy contact of the ink droplet to the substrate.

In order to achieve a stable state with high entropy, A- (green spheres) and B-NPs (red spheres) preferentially mix evenly with each other (Figure 1b). Therefore, when the ink is mixed via sufficient vortexing (30 s), the A- and B-NPs remain in an evenly mixed and dispersed state. Moreover, the sedimentation of metallic NPs dispersed in the ink is negligible. The metallic NPs dispersed in the ink sink by receiving the buoyancy, $F_{B}=v\left(\tilde{\rho }-\rho \right)g$, due to the difference in mass density with the dispersion medium (water), and at this experience the Stokes drag force, $F_{D}=-3\pi D\eta u_{D}$, due to the sinking speed $u_{D}$. Here, $v=\frac{\pi D^{3}}{6}$ is a volume of NP with a diameter *D*. Since the system corresponds to the Low Reynolds number (Re=ρDuDη≪1), their sinking speed is determined by the balance of the buoyancy and the Stokes drag force, resulting in $u_{D}=\frac{v(\tilde{\rho }-\rho )g}{3\pi D\eta }$. Where η is the viscosity of water. Metallic NPs also have a speed $u_{B}=\sqrt{\frac{3k_{B}T}{v\tilde{\rho }}}$ due to Brownian motion associated with thermal fluctuations, and thus the sedimentation is negligible due to $|\frac{u_{D}}{u_{B}}|\ll 1$. Here, $k_{B}$ and *T* are Boltzmann constant and absolute temperature, respectively. In other words, the dispersion state of the metallic NPs in the ink is maintained uniformly during the fabrication process.

The 3D colloidal assembly was achieved by a stamping process consisting of three steps, Accumulation–Contact–Detachment (Figure 1c). Adjusting the accumulation time ($t_{A}$) in the first step is crucial for precise volume control of the 3D colloidal assembly where the NFP tip is waiting at a sufficiently far distance from the substrate. In this step, the water of the ink evaporates continuously at the NFP tip, and NPs are gradually accumulated on the tip. The accumulated NPs are transported on the substrate through the capillary bridge which connects the tip and substrate in the contact step for 1 s. In the last detachment step, the substrate moves rapidly downward and the capillary bridge is disconnected leaving NPs on the substrate. The NPs stacked on the substrate exhibit interesting 3D colloidal assembled structures which are dependent on accumulation time, inner diameter of the NFP tip, and mixing state of A- and B-NPs.

### 3.2. Fabrication Mechanism of a Femtoliter Level of 3D Colloidal Assembly

In order to more precisely control the 3D colloidal assembly at the femtoliter level using the nano-fountain pen (NFP), an analytic solution from the evaporation of the ink solvent was found. We assumed the shape of a protruding water droplet to be a hemisphere with $R_{1}$ to simplify the calculation when the ink is filled in the cartridge higher than the critical height ($H_{C}$) as explained in Figure 1b. The mass loss of the solvent due to evaporation occurring on the surface of the droplet can be calculated by the diffusion equation for solvent expressed as $\frac{\partial \tilde{c}}{\partial t}=C_{D}\nabla^{2}\tilde{c}$, and the mass loss generated here is filled with the ink in the cartridge as soon as it is supplied so that the size of the droplet always remains the same (blue arrow in Figure 2a). Here, $C_{D}∼2.6\times 10^{-5}\;m^{2}/s$ is the diffusion constant and $\tilde{c}$ is the vapor concentration for water [31]. In addition, due to the very small droplet size and the relatively large reservoir size, the diffusion of the solvent quickly reaches a steady condition and the diffusion equation becomes the Laplace equation expressed as $C_{D}\nabla^{2}\tilde{c}=0$. Since the concentration function must be continuous, the vapor concentration of the solvent near the droplet surface becomes saturated (defined as ${\tilde{c}}_{s}∼19.96\;g/m^{3}$) for a given temperature, and the vapor concentration corresponds to the relative humidity of the reservoir (defined as ${\tilde{c}}_{0}∼7.99\;g/m^{3}$) when the distance is far enough away. Under these conditions, the distribution of vapor concentration can be represented by $\tilde{c}={\tilde{c}}_{0}-\left({\tilde{c}}_{s}-{\tilde{c}}_{0}\right)R_{1}/r$ as shown in Figure 2b, and the flux of the solvent by evaporation can be expressed as $\vec{J}=-C_{D}\nabla \tilde{c}$ from the spatial gradient of the vapor concentration. By integrating the flux passing through the closed surface surrounding the droplet, the mass loss rate of the solvent ($\dot{M}=dM/dt$) due to evaporation can be obtained as $\dot{M}=1/2\times \oint \vec{J}\cdot d\vec{a}$ where the one-half multiplied before the integral is due to only the hemisphere surface contributing to evaporation. Consequently, the volume loss rate of the ink solvent can be calculated as $\dot{V}=\dot{M}/\rho$ (Figure 2c).

The picoliter levels of volume loss per unit of time occurring at the NFP tip leads results in a volume change of the 3D colloidal assembly (Figure 2a). The volume loss of the ink solvent is proportional to the evaporation time ($t_{e}$) in the accumulation step which is now defined as the term accumulation time ($t_{A}$). The NPs are accumulated continuously at the NFP tip by evaporation, and then the accumulated NPs volume should also increase gradually. The concentration of NPs dispersed in the ink was c=5mg/mL, and the mass of accumulated NPs ($M_{A}=c\dot{V}t_{A}$) was yielded to the accumulated volume of NPs ($V_{A}=M_{A}/\tilde{\rho }$) to approximate 1 fL for $t_{A}=5\;s$ and the mass density of ${\mathrm{AuNP}}_{77}$, $\tilde{\rho }=19.32\;g/{\mathrm{cm}}^{3}$. This estimation coincides with the result as shown in Figure 2d. The volume of the 3D colloidal assembly is monotonically increased according to the accumulation time. Interestingly, as the accumulation time increases, the empty space of the half-torus is gradually filled and the shape of the 3D colloidal assembly is changed to that of a short pillar. This confirms that the shape, as well as the volume of the 3D colloidal assembly, can be changed by adjusting the accumulation time. The inner diameter size of the NFP tip also influences the volume of the 3D colloidal assembly, with the results shown in Figure 2e,f exhibiting that the larger inner diameter forms a larger volume for the same accumulation time. Moreover, the ink mixed with A- and B-NPs also forms the 3D colloidal assembly well (Figure 2g) where the bright and dark NPs represent ${\mathrm{AuNP}}_{77}$ and ${\mathrm{AgNP}}_{71}$, respectively. The packing structures of the 3D colloidal assemblies as shown in the bottom images of Figure 2e–g seem like random closed packing and this is analyzed later in greater detail.

### 3.3. Packing Structures of 3D Colloidal Assembly

The characteristics of the packing structure were found from the position of the center of mass of the NPs in the 3D colloidal assembly. Each NP can be identified in the SEM image, and their center of mass (${\vec{r}}_{i}$) was manually located which is marked in red dots (Figure 3a), and the center positions are plotted as shown in Figure 3b, and using these, we found the distances between two NPs ($r_{ij}=\left|{\vec{r}}_{j}-{\vec{r}}_{i}\right|$, for i≠j). We found the number of the first nearest neighbors from a certain NP as marked in red and blue circles.

The 3D colloidal assembly exhibited a random close packing. The distribution of distance (black dots) and their cumulative percent (red solid line) were counted for the distance interval of $\Delta r_{ij}=10\;\mathrm{nm}$ (Figure 3c). Since the measuring area is finite, the tail of the distribution of distance converges to zero. To find the packing distance, we focused on the distribution for $r_{ij}<0.2\;\mu m$ with smoothing (red dots of Figure 3d). The result showed two peaks at 73 nm and 141 nm, which correspond with the distances for the first (FNNs) and third nearest neighbors (TNNs), while the peak corresponding with the second nearest neighbors (SNNs) is absent. Since the centers do not lie in one plane, the distance of the FNN measures slightly less than 77 nm. In the case of the TNNs, the distance is slightly less than twice that of FNNs, meaning that the particles in the TNNs are not lined up exactly, but rather slightly deviated. The absence of the peak for SNNs also indicates the structure is not hexatic packing and we investigated the number of FNNs and their ratio regarding the number (Figure 3e). The number of FNNs was counted for each NP located within a certain radius that is set to 1.5 times the distance for FNNs. The highest ratio occurred at 6, 5, and 7 follows suggesting that the structure is not hexatic. Therefore, we confirmed that the 3D colloidal assembly is random close packing through comprehensive structural analysis. For random close packing, the volume fraction of the 3D colloidal assembly is α≈0.64 [32], and thus the 3D colloidal assembly volume ($V_{3D}$) and accumulated volume can be expressed as $\alpha V_{3D}=V_{A}$. In the cases of 1, 2, and 3, the number is influenced by the NPs’ position where the NPs are located near the border of the measuring area while the 8 and 9 are observed because of the NPs located beneath the outer layer.

### 3.4. Optical Properties Regarding the Structural Changes of 3D Colloidal Assembly

Clusters of metallic NPs created by the 3D colloidal assembly can be used for developing optical metamaterials via wavelength-selective light scattering due to localized surface plasmon resonance (LSPR) [18,19,20,21]. In order to tune LSPR, the structure of the cluster needs to be controlled, since the packing structure and mixing of metallic NPs are responsible for the LSPR change. The advantage of using NFP is that the mixing and dispersion state of the NPs dispersed in the ink determines the packing characteristics of the 3D colloidal assembly. It is possible to study the optical properties of optical metamaterials more easily by tuning and fabricating 3D colloidal assemblies of single or binary configurations.

The strategy for tuning LSPR was approached by changing the composition of the NPs and changing the packing structure. For changing the composition of the NPs, the NPs composed of gold (G and g), silver (S and s), and platinum (P) are used, and the large and small NPs are also adopted for changing packing structure (Figure 4a). In order to tune the dispersion state of the 3D colloidal assembly, it is important to adjust the mixing ratio of the ink regarding the NP size. The dispersion state of the 3D colloidal assembly composed of two types of A-NPs and B-NPs is largely determined by the size difference between them. We considered two major states: (I) The size of A-NPs and B-NPs are similar, and (II) A-NP is approximately four-times larger than B-NP. Firstly, we set the conditions so that B-NPs enclose A-NPs or are enclosed by A-NPs for 3D colloidal assembly, and the number density ratio was estimated from the radii of A-NPs and B-NPs, $R_{A}$ and $R_{B}$. The spherical surface area $S=4\pi {\left(R_{A}+R_{B}\right)}^{2}$, and the approximate number of B-NPs needed to wrap A-NP in closed packing is $\frac{1}{2}\frac{S}{\pi R_{B}^{2}}$. Here, $\frac{1}{2}$ was multiplied as when A-NPs are enclosed, they share B-NPs. Therefore, the number density ratio of 10:1 and 1:10 for case (I) and 1:50 for case (II) is sufficient for the enclosing condition in 3D colloidal assembly. Considering these points, we finally chose the number density ratio of A-NPs and B-NPs, ($n_{A}:n_{B}$) as follows: (1:0, 10:1, 1:1, 1:10, 0:1) for case (I) and (1:0, 1:1, 1:50, 0:1) for case (II). The packing structure changes following the yellow arrow for case (I) and the red arrow for case (II) as shown in Figure 4b. We note that approximately 1–50 nanoparticles are dispersed in a 1μm cube box in the ink in a state that satisfies the experimental conditions described previously.

The spectrum of scattered light is changed by the 3D colloidal assembly of various structures made using NFP. For case (I) of ${\mathrm{PtNP}}_{72}$ and ${\mathrm{AgNP}}_{71}$, the number density ratio ($n_{P}:n_{G}$) is (1:0, 10:1, 1:1, 1:10, 0:1), their scattered light spectrum changes (Figure 4c), and the RBG value changes as shown in Figure 4d when these spectra are converted using the CIE 1931 color space. The 3D colloidal assemblies for different four kinds of NPs as shown in Figure 4a exhibited color changes according to the mixing ratio (Figure 4e). The dark star (P) represents the spectrum obtained from the single-component 3D colloidal assembly consisting of PtNPs. The red stars exhibit the spectra measured by the single-component 3D colloidal assembly for ${\mathrm{AuNP}}_{77}$ (G), ${\mathrm{AuNP}}_{20}$ (g), ${\mathrm{AgNP}}_{71}$(S), and ${\mathrm{AgNP}}_{26}$(s). The yellow and red arrows in Figure 4e indicate the structure change as shown in Figure 4b. Their color changes occurred with the signals of two single-component 3D colloidal assemblies. We anticipate that by carefully using metal nanoparticles whose scattering spectra occur in parts distinct from the color map, a wide color distribution between the two parts can be achieved. Moreover, it is expected that the fabrication of a simple 3D colloidal assembly using NFP and the adjustment of the mixing composition ratio will be applicable to the study of micrometer-scale optical metamaterials.

## 4. Conclusions

We investigated the fabrication and principles of a 3D colloidal assembly inspired by a nano-fountain pen (NFP). When the inner diameter of the NFP tip was reduced to a micrometer-size scale, 1 pL of water per second was evaporated with the ink solvent generated at the tip. This evaporation caused the metal nanoparticles (NPs) dispersed in the ink to accumulate on the NFP tip. The accumulated volume increased in proportion to the accumulation time with a rate of increase of 1 fL per second. Therefore, through the accumulation phenomenon and the stamping process, a 3D colloidal assembly with a volume of 1 fL could be fabricated on a Si wafer substrate. Depending on the accumulation time, the shape of the manufactured 3D colloidal assembly changed from half-torus to short-pillar. Through examining the location of the center of mass of the NPs constituting the 3D colloidal assembly, the structure was found to have a random close packing. In addition, by changing the mixing of metal nanoparticles dispersed in the ink, 3D colloidal assemblies of various sizes and material compositions were fabricated, and the possibility of localized surface plasmon resonance tuning was confirmed. It is expected that the fabrication mechanism presented here, as well as the simple binary cluster fabrication method, can be used not only for research on various combinations of optical metamaterials but also for the development of optical sensor devices and nanocatalysis using this structure [33,34,35].

## Figures and Tables

**Figure 1 nanomaterials-13-02403-f001:**
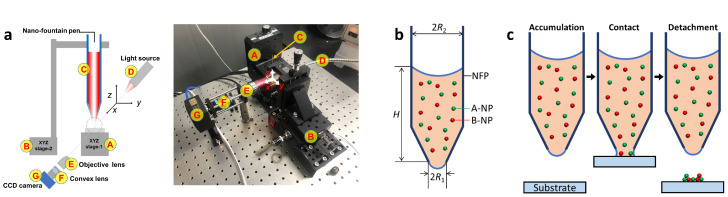
Fabrication of 3D colloidal assembly using NFP. (**a**) Experimental setup for 3D colloidal assembly: schematic (left) and picture (right). (**b**) Schematic of NFP and NPs dispersed in the ink. (**c**) Stamping process to form the 3D colloidal assembly.

**Figure 2 nanomaterials-13-02403-f002:**
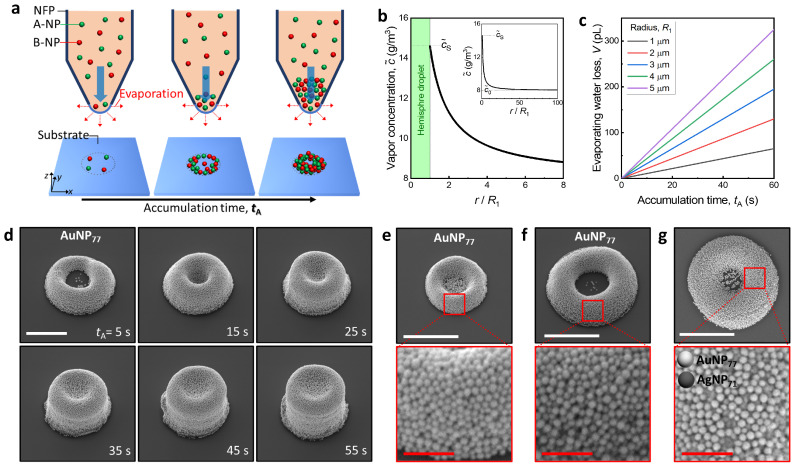
Volume and shape depending on evaporation. (**a**) Volume increment of 3D colloidal assembly regarding the accumulation time, $t_{A}$. (**b**) Distribution of vapor concentration induced by evaporation at the NFP tip. (**c**) Evaporation-induced mass loss of water according to the tip inner radius. (**d**) Scanning electron microscope (SEM) images of 3D colloidal assemblies consisting of ${\mathrm{AuNP}}_{77}$. The scale bar represents 2μm. SEM images of ${\mathrm{AuNP}}_{77}$ assemblies for $t_{A}=10\;s$: (**e**) $R_{1}∼3\;\mu m$ and (**f**) $R_{1}∼5\;\mu m$. (**g**) Binary assembly consisting of ${\mathrm{AuNP}}_{77}$ (bright particles) and ${\mathrm{AgNP}}_{71}$ (dark particles). In Figure 2e–g, the scale bars colored white (top) and red (bottom) represent 3μm and 0.5μm, respectively.

**Figure 3 nanomaterials-13-02403-f003:**
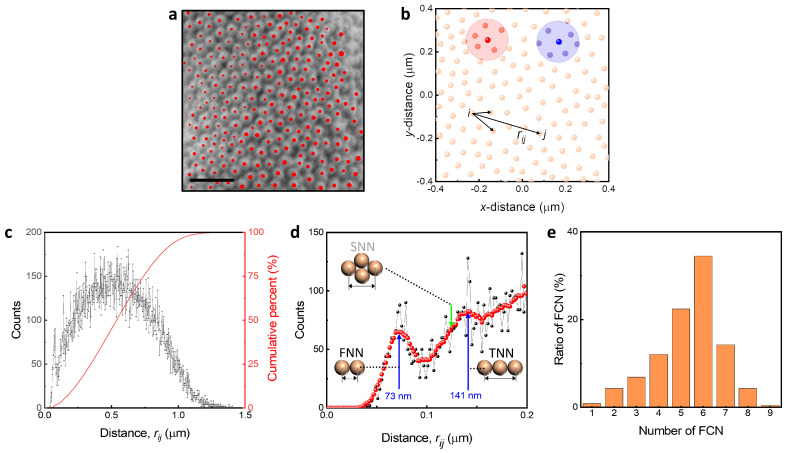
Packing structure of 3D colloidal assembly. (**a**) Center of mass of ${\mathrm{AuNP}}_{77}$. The scale bar represents 0.3μm. (**b**) Characterization of packing structure using the distance between two NPs ($r_{ij}$) and the number of first nearest neighbors (FNNs). (**c**) Distribution of $r_{ij}$. The counts represent the number of $r_{ij}$ existing within distance increment for each $\Delta r_{ij}=10\;\mathrm{nm}$. (**d**) Packing distance of 3D colloidal assembly. (**e**) Ratio of FNNs. The number of FNNs was found by counting the NPs existing within 1.5 times the FNN distance.

**Figure 4 nanomaterials-13-02403-f004:**
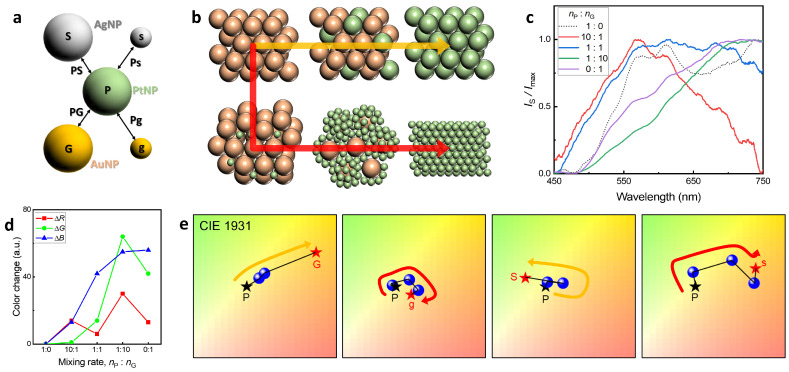
Optical property of the 3D colloidal assembly regarding the diverse structures and mixing compositions. Schematics for (**a**) Mixing combinations of metallic NPs and their (**b**) Packing structures of 3D colloidal assemblies regarding composition and size. The colors represent the composition of NP. The yellow and red arrows show the changes in packing structures when the mixed NP sizes are similar and different, respectively. (**c**) Light scattering spectra of 3D colloidal assemblies consisting of ${\mathrm{AuNP}}_{77}$ and ${\mathrm{AuNP}}_{20}$. (**d**) RGB color changes depending on the mixing ratio. (**e**) Color shift of diverse 3D colloidal assemblies on the CIE 1931 color space. The yellow and red arrows represent packing structure changes as described in Figure 4b.

## Data Availability

The data presented in this study are available on request from the first author.

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
