# Peer review of "Fountain Pen-Inspired 3D Colloidal Assembly, Consisting of Metallic Nanoparticles on a Femtoliter Scale"

_nanomaterials, 2023, doi:10.3390/nano13172403_

Round 1

Reviewer 1 Report

Jin-Woo et al. developed the fountain pen-inspired 3D colloidal metallic nanoparticles on a femtoliter scale. The technology is important way to prepare metallic nanoparticles assembly. The paper is well organized. I suggest the publication of the work after some changes.

The TEM images for these Au, Ag, Pt nanoparticles assembly should be provided, which is important in the nanomaterials. The similar work of 10.34133/research.0018 should be cited. For example, the Au(111) facets should be found. Furthermore, the size of colloidal metallic nanoparticles in the parent solution is same with that in the final nanoparticles assembly or not. Moreover, what is application of these Au, Ag, Pt nanoparticles assembly, e.g. nanocatalysis, e.g. 10.1038/s42004-023-00817-5, 10.1002/tcr.202100001.

Reviewer 2 Report

The paper is dedicated to the preparation of metamaterials with optical properties. The 3D colloidal assembly was obtained with a fountain pen methodology. The mechanism of nanoparticles assembly was carefully studied and discussed and the nanoparticles assemblies were characterized by spectroscopic and microscopic analysis. The study is interesting, well written and can pave the route for the development of nanostructured materials with controlled physico-chemical properties.
